# Uncertainty Estimation and Calibration in nnU-Net-Based Ischemic Stroke Lesion Segmentation

**Ewout Heylen**[1,2] (iD)                              EWOUT.HEYLEN@KULEUVEN.BE

[1]*Department of Electrical Engineering (ESAT), KU Leuven, Leuven, Belgium*

[2]*Stanford Stroke Center, Palo Alto, CA, USA*

**Jelle Demeestere**[3,4] (iD)

[3]*Department of Neurology, University Hospitals Leuven, Leuven, Belgium*

[4]*Division of Experimental Neurology, Department of Neurosciences, KU Leuven, Leuven, Belgium*

**Anke Wouters**[4] (iD)

**Pierre Seners**[5,6] (iD)

[5]*Department of Neurology, Hôpital Fondation A. de Rothschild, Paris, France*

[6]*INSERM U1266, Institut de Psychiatrie et Neurosciences de Paris (IPNP), Paris, France*

**Soren Christensen**[2] (iD)

**Nicole Yuen**[2] (iD)

**Michael Mlynash**[2] (iD)

**Stephanie Kemp**[2]

**Jeremy J. Heit**[2] (iD)

**Gregory W. Albers**[2] (iD)

**Seena Dehkharghani**[7] (iD)

[7]*Department of Radiology, Stanford University School of Medicine, Stanford, CA, USA*

**Robin Lemmens**[3,4] (iD)

**Maarten Lansberg**[2] (iD)

**Frederik Maes**[1] (iD)

**Editors:** Accepted for publication at MIDL 2025

## Abstract

nnU-Net has become widely recognized as a state-of-the-art semantic segmentation framework. However, deep learning models are often poorly calibrated, resulting in unreliable probability estimates. Additionally, they lack meaningful uncertainty quantification. We trained an nnU-Net model to segment ischemic stroke lesions on acute-phase Diffusion-Weighted Imaging (DWI) MRI and applied a Bayesian posterior sampling approach to estimate uncertainty and improve model calibration. Our findings show that the Bayesian posterior sampling approach yields better calibration compared to a conventional nnU-Net, while providing uncertainty estimates and maintaining comparable segmentation performance.

**Keywords:** nnU-Net, Model Calibration, Uncertainty Estimation, Ischemic Stroke.

## 1. Introduction

Deep learning (DL) models have been applied to various medical imaging segmentation tasks. nnU-Net has become widely recognized as a state-of-the-art semantic segmentation

framework (Isensee et al., 2021). Although segmentation is often treated as a binary classification problem, the underlying probability predictions and accompanying uncertainty can potentially provide a more comprehensive evaluation of the lesion extent. While DL models have become more effective, they are often poorly calibrated. Model capacity, patch size, batch normalization, weight decay, and soft dice loss have been shown to influence model calibration. Various techniques have been described to calibrate an AI model after training, such as scaling methods, binning methods, and ensembling techniques (Guo et al., 2017; Mehrtash et al., 2020; Wang et al., 2023). Besides, uncertainty estimation helps assess the reliability of the model's predictions and can be classified as Bayesian (e.g., Monte Carlo dropout) or non-Bayesian (e.g., ensembling) methods (Mehrtash et al., 2020).

## 2. Methods

A conventional nnU-Net model was trained for ischemic stroke lesion segmentation on b1000 DWI images of patients with an anterior circulation large vessel occlusion admitted to a comprehensive stroke center for consideration of endovascular therapy (CRISP2 dataset) (Wouters et al., 2024). Bias field correction using the N4 algorithm was applied prior to training (Tustison et al., 2010). The DL model was trained with five-fold cross-validation using nnU-Net's default settings, including 1000 training epochs. The model's performance and calibration were evaluated on an external test set consisting of b1000 DWI images from the University Hospitals Leuven, Belgium. The softmax probabilities across the five folds were averaged to generate the final prediction. A binary segmentation map was obtained by applying a threshold of 0.5. A quantitative evaluation was performed using the Dice similarity coefficient (DSC) and absolute volume difference (AVD).

A second training scheme was defined to enable Bayesian uncertainty estimation, similar to a previously proposed method for capturing both local and global uncertainty of the weights (Zhao et al., 2022; Vorberg et al., 2024). However, to facilitate integration with the nnU-Net framework, the default nnU-Net settings were used, except for the number of training epochs. To mimic a cyclic learning rate scheme, five folds were trained for 200 epochs, and for each fold, the last 10 checkpoints were sampled. The choice of 200 epochs was based on the learning curves of the conventional nnU-Net, where the validation loss plateaued around epoch 200 while the training loss continued to decrease. Final probability maps were obtained by averaging the predictions from all collected checkpoints (n = 50). A threshold of 0.5 was applied to obtain a binary segmentation map. Voxel-wise model (epistemic) uncertainty was evaluated by computing the variance across the 50 checkpoints.

## 3. Results

The training set included 197 images with a median lesion volume of 17.0 ml (IQR: 3.1 - 42.3 ml). The external test set included 75 images with a median lesion volume of 15.1 ml (IQR: 4.8 - 37.8 ml). The performance metrics on the test set were similar for the conventional model and the Bayesian model, with a mean DSC of 0.60 for both models and a mean AVD of 9.6 ml and 9.8 ml for the conventional model and the Bayesian model, respectively. A table containing more detailed metrics is provided in the appendix (Table 1). Uncertainty estimates and probability maps of the Bayesian model are shown in Figure 1.

The Maximum Calibration Error (MCE) and Expected Calibration Error (ECE) for the conventional model were 43.4% and 1.1%, respectively. The Bayesian model achieved improved calibration, with an MCE of 29.5% and an ECE of 1.0%. However, due to the substantial class imbalance between lesion and non-lesion voxels, the ECE values may be biased and not fully reflect calibration performance. The calibration curves for both models are shown in the appendix (Figure 2).

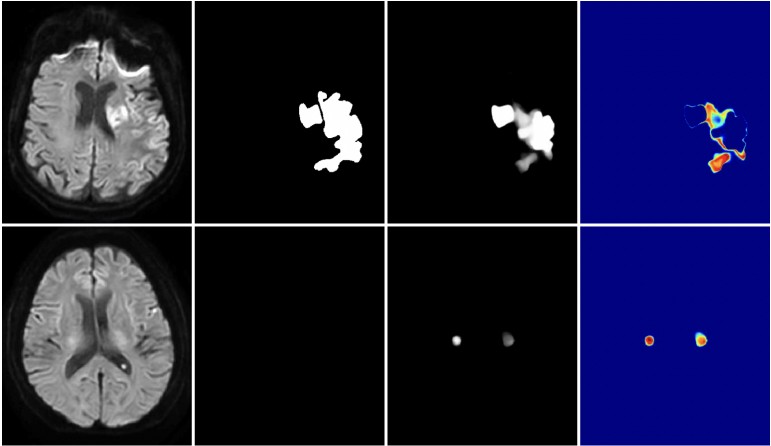

Figure 1: Uncertainty estimates and probability maps of the Bayesian model. Each row corresponds to a different patient. Columns (from left to right): (1) DWI image, (2) ground truth, (3) probability map, and (4) uncertainty map. Red indicates higher uncertainty, blue indicates lower uncertainty.

## 4. Discussion

Uncertainty estimation and model calibration help assess the reliability and explainability of segmentation models, allowing for more nuanced interpretations than binary segmentation maps, and facilitating the handling of domain shift. The proposed training scheme can improve calibration and is easy to implement with the nnU-Net framework. Additionally, it enables uncertainty estimation, as shown in Figure 1. In the first row, the uncertainty map mostly corresponds to the probability map. However, between the two areas with high probabilities, there is a region showing similar probability values, while the uncertainty map indicates varying levels of uncertainty. In the second row, the probability map incorrectly highlights two areas. The region with the highest probabilities corresponds to the highest uncertainties, suggesting that predicted probabilities and epistemic uncertainty capture different aspects of the model's output. In terms of performance, the DSC and AVD of the Bayesian model were comparable to those of the conventional nnU-Net. However, averaging predictions across all checkpoints leads to higher computational costs at inference time.

## 5. Conclusion

We applied an easy-to-implement training scheme compatible with the nnU-Net framework to improve the calibration of an ischemic stroke lesion segmentation model, while also enabling uncertainty estimation and maintaining comparable segmentation performance.

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

## Appendix

Table 1: Performance metrics on the test set: Dice similarity coefficient (DSC) and absolute volume difference (AVD). Statistical measures: standard deviation (SD) and interquartile range (IQR).

| Performance metric | Conventional model | Bayesian model |
|---|---|---|
| Mean (SD) DSC | 0.60 (0.24) | 0.60 (0.25) |
| Median (IQR) DSC | 0.67 (0.50 - 0.78) | 0.67 (0.49 - 0.79) |
| Mean (SD) AVD (ml) | 9.6 (17.1) | 9.8 (18.4) |
| Median (IQR) AVD (ml) | 2.7 (0.6 - 9.5) | 3.5 (0.7 - 9.3) |

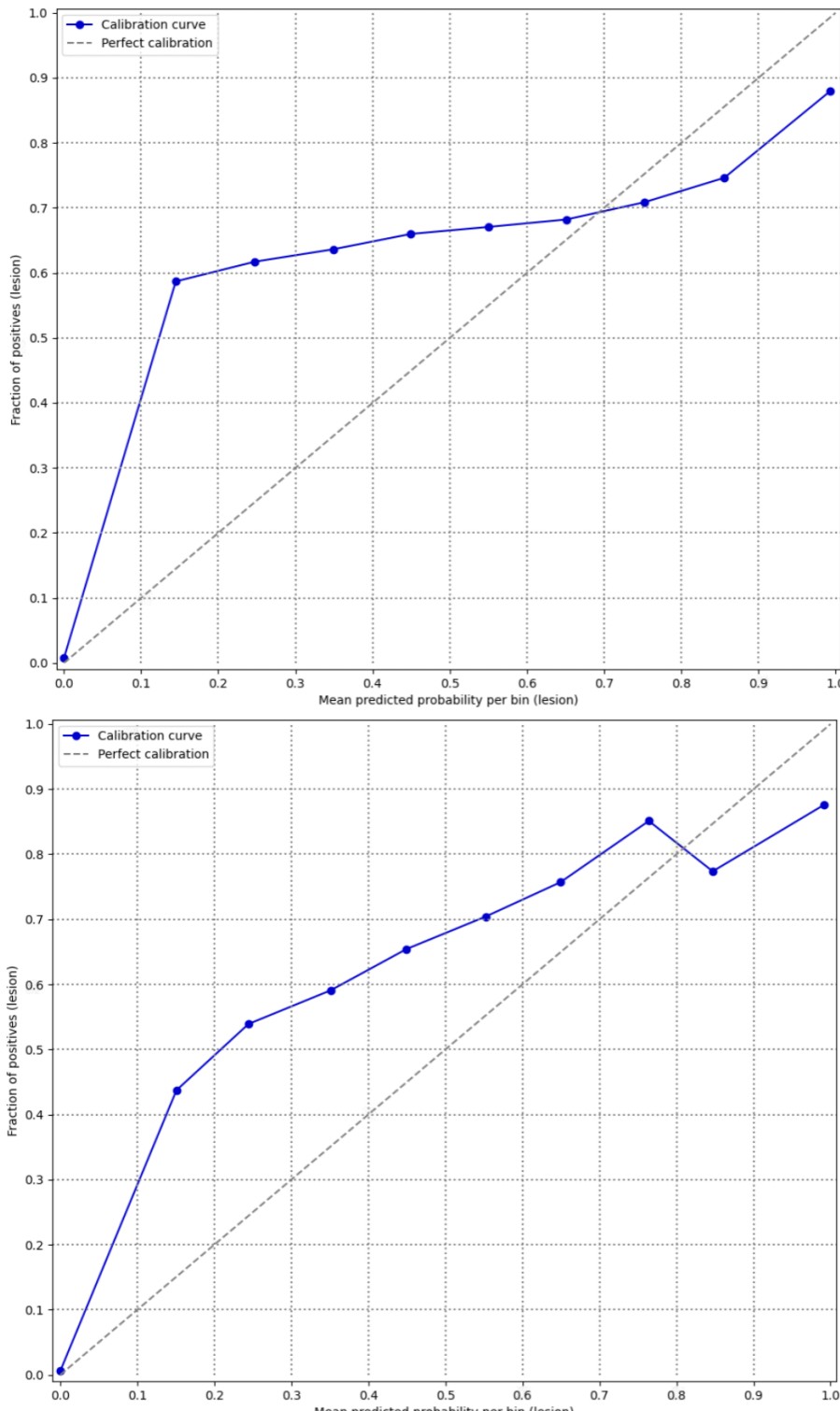

Figure 2: Calibration curves for the conventional (top) and Bayesian (bottom) models. The x-coordinate represents the mean predicted probability for each bin, computed for the positive class (i.e., lesion).

