# OpenReview forum: "Uncertainty Estimation and Calibration in nnU-Net-Based Ischemic Stroke Lesion Segmentation"
_MIDL.io/2025/Short_Papers — MIDL 2025 - Short Papers_

### Official Review · Reviewer_XcXc · 2025-04-28

**Rating:** 5
**Confidence:** 5

**Summary:**

This paper presents a training scheme compatible with the nnU-Net framework to improve the calibration of an ischemic stroke lesion segmentation model. This method also enables uncertainty quantification. Propose method is applied to ischemic stroke lesion segmentation

**Strengths:**

This paper is clear and well-written. It deals with the important problem of uncertainty quantification and model calibration. DL models are often poorly calibrated. Proposed method is very interesting. The authors trained an nnU-Net model to segment ischemic stroke lesions on acute-phase Diffusion-Weighted Imaging (DWI) MRI and applied a Bayesian posterior sampling approach to estimate uncertainty and improve model calibration.

**Weaknesses:**

There is no specific weakness for this short paper.

---

### Decision · Program_Chairs · 2025-05-01

Accept